# Atomic overlayer of permeable microporous cuprous oxide on palladium promotes hydrogenation catalysis

Kunlong Liu[1,8], Lizhi Jiang[1,2,8], Wugen Huang[3,4,8], Guozhen Zhu[5], Yue-Jiao Zhang[1], Chaofa Xu[1], Ruixuan Qin[1], Pengxin Liu[1], Chengyi Hu[1], Jingjuan Wang[1], Jian-Feng Li[1], Fan Yang[3,6✉], Gang Fu[1,7✉] & Nanfeng Zheng[1,7✉]

The interfacial sites of metal-support interface have been considered to be limited to the atomic region of metal/support perimeter, despite their high importance in catalysis. By using single-crystal surface and nanocrystal as model catalysts, we now demonstrate that the overgrowth of atomic-thick $Cu_2O$ on metal readily creates a two-dimensional (2D) microporous interface with Pd to enhance the hydrogenation catalysis. With the hydrogenation confined within the 2D $Cu_2O$/Pd interface, the catalyst exhibits outstanding activity and selectivity in the semi-hydrogenation of alkynes. Alloying Cu(0) with Pd under the overlayer is the major contributor to the enhanced activity due to the electronic modulation to weaken the H adsorption. Moreover, the boundary or defective sites on the $Cu_2O$ overlayer can be passivated by terminal alkynes, reinforcing the chemical stability of $Cu_2O$ and thus the catalytic stability toward hydrogenation. The deep understanding allows us to extend the interfacial sites far beyond the metal/support perimeter and provide new vectors for catalyst optimization through 2D interface interaction.

[1] State Key Laboratory for Physical Chemistry of Solid Surfaces, Collaborative Innovation Center of Chemistry for Energy Materials, and National & Local Joint Engineering Research Center for Preparation Technology of Nanomaterials, College of Chemistry and Chemical Engineering, Xiamen University, Xiamen 361005, China. [2] The Straits Institute of Flexible Electronics (SIFE, Future Technologies), Fujian Normal University, Fuzhou 350117, China. [3] State Key Laboratory of Catalysis, Dalian Institute of Chemical Physics, Chinese Academy of Sciences, Dalian 116023, China. [4] University of Chinese Academy of Sciences, Beijing 100049, China. [5] Department of Mechanical Engineering and Manitoba Institute of Materials, University of Manitoba, Winnipeg, MB R3T 5V6, Canada. [6] School of Physical Science and Technology, ShanghaiTech University, Shanghai 201210, China. [7] Innovation Laboratory for Sciences and Technologies of Energy Materials of Fujian Province (IKKEM), Xiamen 361102, China. [8] These authors contributed equally: Kunlong Liu, Lizhi Jiang, Wugen Huang. ✉email: fyang@shanghaitech.edu.cn; gfu@xmu.edu.cn; nfzheng@xmu.edu

Supported metal catalysts are an important category of heterogeneous catalysts that are widely applied in the chemical industry[1–4]. When catalytic metal species are deposited onto supports or decorated by supports, a two-dimensional (2D) metal-support interface is naturally formed[5–9]. It has been well demonstrated that the interface can effectively modulate the adsorption behavior of reactants, intermediates, or products on the different chemical components of the interface, thus providing the opportunity to break the linear scaling relationship for enhancing the catalytic performance[10–12]. In most cases, oxide supports with dense structures have no accessible porosity by small molecules so the catalytically active sites have to be limited to the atomic region of 1D metal/support perimeter[13–16]. Under such a situation, it is very unfavorable for the design of supported catalysts with optimized usage of noble metals. It is hence highly desirable to explore the feasibility of applying porous oxide supports to extend the catalytic interfacial sites beyond the metal/support boundary.

We now demonstrate the in situ formation of atomic-thick microporous 2D $Cu_2O$ overlayer on PdCu alloy to achieve an unprecedentedly simultaneous improvement in activity, selectivity, and stability in catalytic hydrogenation. In this work, high-surface nanosheets (NSs) and single-crystal surfaces with PdCu alloy covered by an atomic-thick microporous $Cu_2O$ overlayer were fabricated and applied as model catalysts to reveal the formation and excellent catalytic performance of 2D $Cu_2O$/Pd catalytic interface. Comprehensive studies demonstrate that the atomic-thick 2D $Cu_2O$ overlayer allows the access of $H_2$ onto the PdCu alloy underneath, which supplies activated H species for hydrogenation occurring on the 2D $Cu_2O$/Pd catalytic interface. In addition, the PdCu alloy underneath the $Cu_2O$ overlayer creates a favorable electronic structure for enhancing catalytic hydrogenation. More importantly, the reaction of terminal alkynes with the boundary or defective sites of the $Cu_2O$ overlayer would build up its robustness against $H_2$ reduction during hydrogenation. The deep understanding allows us to prepare practical catalysts with remarkable performance enhancement in the activity, selectivity and stability toward the semi-hydrogenation of a wide range of alkynes under mild conditions.

## Results

### Fabrication and characterization of PdCu@Cu₂O core-shell nanocatalysts

The 2D $Cu_2O$ monolayer has a microporous framework with pore opening of ~5.5 Å, which can restrict the passage of medium-sized molecules[17]. More importantly, the formation of 2D microporous $Cu_2O$ overlayer on single-crystal surface of noble metal has been well documented in the literature[18–20]. Experimentally, to create Pd catalysts with 2D microporous $Cu_2O$ overlayer, Pd@Cu core-shell NSs were first prepared by depositing Cu on the premade Pd NSs with a thickness ~5-atomic layer (Fig. 1a and Supplementary Fig. 1) via a hydride-induced-reduction strategy[21,22]. The as-prepared Pd@Cu NSs were then exposed in air at 30 °C for 48 h to prepare the PdCu@Cu₂O core-shell NSs[23,24]. As revealed by transmission electron microscopy (TEM), the hexagonal shape of Pd NSs was deformed after the Cu deposition (Fig. 1b and Supplementary Fig. 2). High-angle annular dark-field scanning transmission electron microscopy (HAADF) and energy-dispersive X-ray elemental mapping (EDS) of PdCu@Cu₂O with Cu/Pd molar ratio of 1 revealed that the growth of Cu was rather uniform on the Pd surface (Fig. 1c and Supplementary Fig. 3). However, the exposure of a small portion of Pd was still revealed by the cyclic voltammetry (CV) (Supplementary Fig. 4).

A series of PdCu@Cu₂O catalysts with different Cu/Pd ratios were prepared and characterized by X-ray absorption spectroscopy (XAS) and X-ray photoelectron spectroscopy (XPS) to investigate the coordination environment and electronic structure of Pd and Cu. In comparison with Pd foil and Pd NSs, the Pd K-edge white line of PdCu@Cu₂O showed a small shift to lower energy for the absorption edge ($E_0$) (Supplementary Fig. 5a, b and Supplementary Table 1). The shift increased with the increased content of Cu, suggesting that Pd in PdCu@Cu₂O was slightly negatively charged with the electron transfer from Cu to Pd. To make the feature of Cu in the samples more evident, the Cu K-edge XANES spectra were subjected to the first derivation[25–27]. As illustrated in Fig. 1d and Supplementary Fig. 5c, the Pd@Cu samples with Cu/Pd ≥ 0.5 displayed two distinct peaks assigned to $Cu^0$ and $Cu^+$, indicating the coexistence of Cu and $Cu_2O$, which was further confirmed by XPS analysis of the Pd@Cu sample with Cu/Pd = 1 (Supplementary Fig. 6 and Supplementary Table 2)[28–30]. In comparison, PdCu@Cu₂O with Cu/Pd < 0.5 only displayed a peak assigned to $Cu^0$, indicating the presence of only metallic Cu in the form of PdCu alloy.

### Catalysis performance of PdCu@Cu₂O core-shell nanocatalysts

The semi-hydrogenation of alkynes was chosen as a model reaction to evaluate the catalytic performance of Pd@Cu with different contents of $Cu^0$ and $Cu^+$[31,32]. As shown in Fig. 1e and Supplementary Fig. 7, the catalytic activity and selectivity of PdCu@Cu₂O in the hydrogenation of phenylacetylene (PhC≡CH) were closely related to the Cu/Pd ratio. It has been well documented that when Pd alloys with Cu, the surface-active Pd sites would be diluted into small domains such that the selectivity can be enhanced often with the suppressed activity[33–35]. As expected, the selectivity toward styrene did increase with the increasing Cu/Pd ratio. Especially, when Cu/Pd ratio was higher than 0.5, all the Pd@Cu catalysts exhibited high selectivity of over 96% to styrene (Fig. 1e and Supplementary Fig. 7). It is worth noting that the catalytic activity increased first and then dropped with the Cu/Pd ratio varying from 0 to 1.5. The optimal ratio of Cu/Pd to achieve the best activity was 1:1 where the total conversion of PhC≡CH was achieved within 80 min at the catalyst/substrate molar ratio of 1:2000. It should be noted that the catalytic performance of the PdCu@Cu₂O (Cu/Pd = 1:1) catalyst was far superior to that of the commercial Lindlar catalyst (Supplementary Fig. 8). Moreover, the catalyst exhibited outstanding stability in the semi-hydrogenation of PhC≡CH. No significant decay in both selectivity and activity was observed even after the catalyst was recycled six times under the same conditions (Supplementary Fig. 9a). In contrast, the catalytic activity of Pd NSs without Cu deposition was gradually decreased in six catalytic cycles (Supplementary Fig. 9b). However, to our big surprise, different from the catalysts collected from the hydrogenation reaction of PhC≡CH, the fresh PdCu@Cu₂O catalyst exhibited a decent hydrogenation activity in the hydrogenation of styrene (Supplementary Fig. 10), indicating that the interaction of PhC≡CH with PdCu@Cu₂O would inhibit the adsorption and hydrogenation of styrene. All these findings suggested that the as-formed catalytically active structure in the hydrogenation of alkynes should be beyond simple PdCu alloy.

### Cu₂O coverage-dependent structure of PdCu bimetallic surface

To understand how the catalytically active structure evolved from the Pd NSs, scanning tunneling microscope (STM) analysis was employed to follow the reaction procedures starting from a single-crystal Pd(111) surface. As shown in Supplementary Fig. 11, when Cu was deposited on Pd(111) with coverage of less than one monolayer (i.e., ~0.96 ML), herringbone dislocation lines were observed on the surface, which is similar to the growth

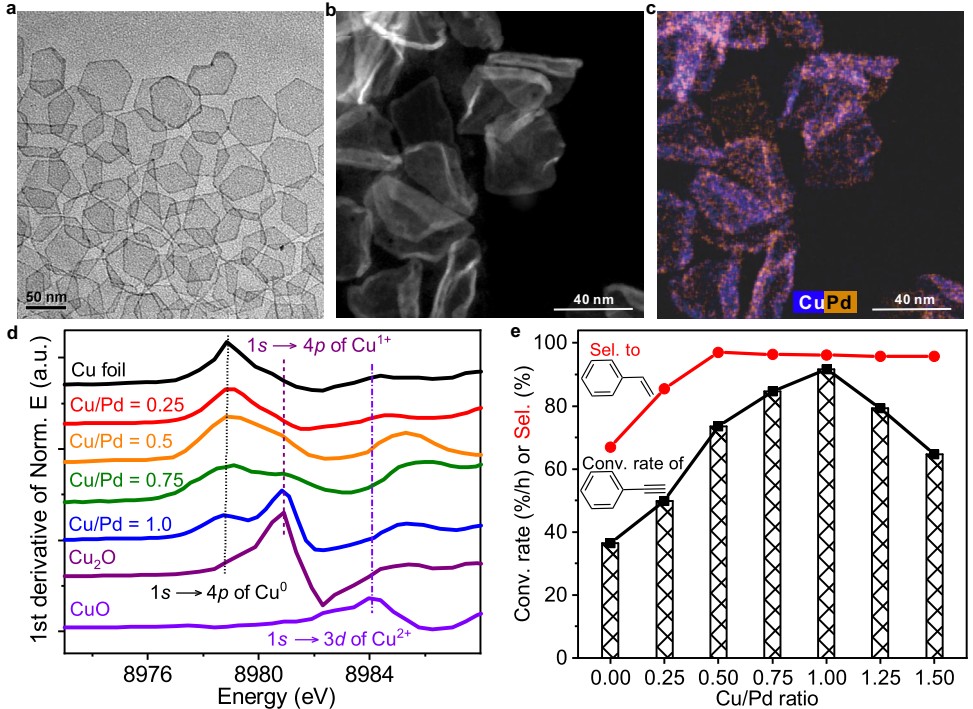

**Fig. 1 Structure characterizations and catalytic performance of PdCu@Cu₂O. a** Representative TEM images of Pd NSs. **b** High-resolution high-angle annular dark-field (HAADF) STEM image of PdCu@Cu₂O (Cu/Pd=1). **c** STEM-EDS elemental mapping of PdCu@Cu₂O (Cu/Pd=1). **d** The first derivatives of Cu K-edge XANES of PdCu@Cu₂O with different Cu/Pd ratio and references. **e** Catalytic performance of semi-hydrogenation of PhC≡CH on PdCu@Cu₂O with different Cu/Pd ratios. Reaction conditions: 10 mL ethanol; 2 μmol Pd; 4 mmol PhC≡CH (1:2,000); $T$ = 303 K; H₂ pressure = 0.1 MPa (PhC≡CH was introduced before H₂).

of Co or Fe thin films on Pt(111)[36,37]. According to density functional theory (DFT) calculations, the deposited Cu atoms with relatively low Cu/Pd ratio preferred to sink into the subsurface (Supplementary Figs. 12 and 13 and Supplementary Table 3), well consistent with the reported theoretical work that Pd on Cu/Pd (1:1) bimetallic alloy was enriched on the outermost layer while Cu was enriched on the second layer[38]. Experimentally, the oxide reduction peak of Cu⁺ in CV was not observed when the deposited Cu/Pd ratio ≤ 0.4 (Supplementary Fig. 14), also indicating the formation of Pd skin in the sample with a low Cu/Pd ratio. However, once the deposited Cu was more than one monolayer, the Cu-on-Pd core-shell structure would be formed (Supplementary Fig. 15), which was further confirmed by the high-sensitivity low energy ion scattering spectra and DFT analysis (Supplementary Figs. 16–18 and Supplementary Table 3). Furthermore, STM results confirmed that, when exposed to O₂, the Cu overlayer on the PdCu surface was easily oxidized to Cu₂O (Fig. 2a and Supplementary Fig. 19)[24,39]. Hence, two kinds of Cu would coexist in the oxidized Pd@Cu samples with the deposition of Cu over one monolayer. While the Cu underneath the Pd surface preferred to keep the metallic state, the surface Cu would be oxidized into +1, well explaining the Cu⁺ signal in the XAS spectra (Fig. 2b and Supplementary Figs. 20 and 21).

Due to their lattice mismatch, the growth of defective oxide overlayer on metal substrate has been commonly observed[19,40]. The rich boundary or defect sites thereon are often proposed to be catalytically active sites. As revealed by STM, when Cu₂O was deposited on Pd(111), rich nanoscale defects were still observed on the oxide overlayer (Supplementary Fig. 19). It was thus reasonable to hypothesize that the boundary between Pd and defective Cu₂O might act as active sites for the semi-hydrogenation of PhC≡CH. However, the hypothesis was disproved by the observation that the PdCu@Cu₂O pretreated

by H₂ displayed a dramatically reduced selectivity toward styrene in the hydrogenation of PhC≡CH. As shown in Fig. 2c and Supplementary Fig. 22, when the conversion of PhC≡CH reached 100%, the selectivity toward styrene was only 76.6% over the H₂-pretreated PdCu@Cu₂O NSs. The STM analysis revealed that the defective Cu₂O on PdCu@Cu₂O was easily reduced to Cu once it was exposed to H₂ at room temperature (Fig. 2d and Supplementary Fig. 23). However, when PhC≡CH was introduced before H₂ treatment, no reduction of defective Cu₂O on PdCu@Cu₂O was observed in $1.0 \times 10^{-6}$ mbar H₂ (Fig. 2e, f and Supplementary Figs. 24 and 25), clearly indicating the reduction of Cu₂O was suppressed by the pretreatment of PhC≡CH. More importantly, once PhC≡CH was introduced to treat the PdCu@Cu₂O (Cu/Pd ≥ 0.5) surface before H₂ feeding, the as-treated surface showed negligible activity for the hydrogenation of styrene (Supplementary Fig. 26). All these results clearly demonstrated that the outstanding selectivity of PdCu@Cu₂O toward styrene in the semi-hydrogenation of PhC≡CH should be attributed to the presence of Cu₂O overlayer on PdCu. The reaction of PhC≡CH at the defective sites of Cu₂O helped to create its robustness against H₂ reduction during hydrogenation.

**Catalytic hydrogenation mechanisms.** To further verify our speculation, in situ Fourier transform infrared spectroscopy (in situ FT-IR) and Raman spectroscopy was performed to monitor the hydrogenation process catalyzed by the PdCu@Cu₂O catalyst prepared from Pd NSs. As shown in Fig. 3a and Supplementary Fig. 27, while the signal of terminal C–H of PhC≡CH disappeared when H₂ treatment was introduced, the signal of the alkynyl-copper (Cu(I)-C≡CPh) complex component still existed even after 80-min H₂ treatment, indicating that the alkynyl group originated from the dissociative adsorption of PhC≡CH should

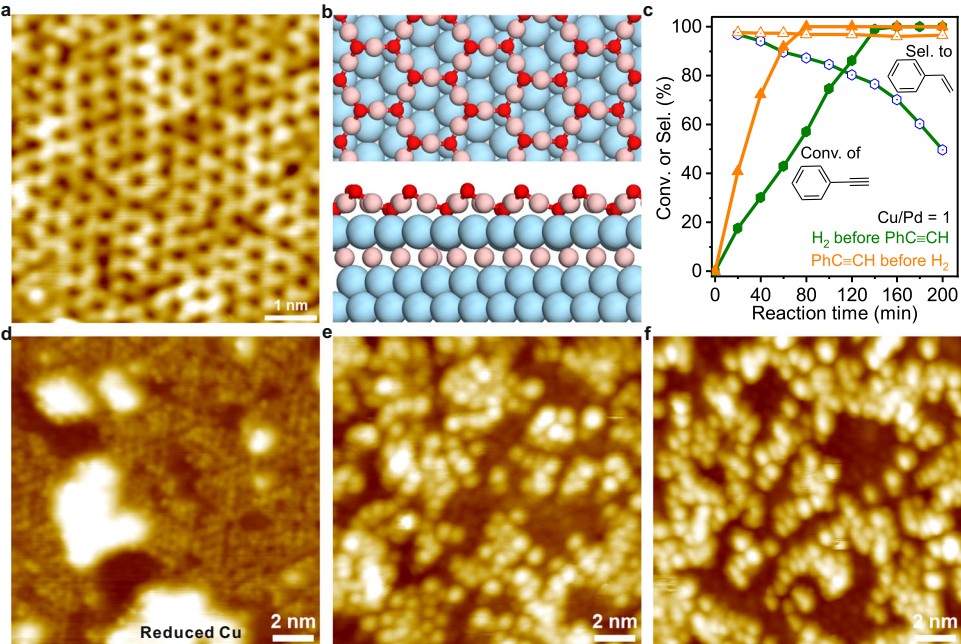

**Fig. 2 Microstructure of PdCu@Cu$_2$O and its redox dynamics. a** STM images of PdCu@Cu$_2$O with >1.0 ML Cu. **b** DFT-optimized structures of PdCu@Cu$_2$O. **c** The difference in catalytic performance of PdCu@Cu$_2$O (Cu/Pd =1) nanosheets caused by the feeding sequence. **d** Exposure of PdCu@Cu$_2$O (~1.0 ML Cu$_2$O) with defects to $1.0 \times 10^{-6}$ mbar H$_2$ at 300 K. **e** PhC≡CH adsorption on surface of PdCu@Cu$_2$O (~1.0 ML Cu$_2$O). **f** Exposure PdCu@Cu$_2$O (~1.0 ML Cu$_2$O) with adsorbed PhC≡CH to $1.0 \times 10^{-6}$ mbar H$_2$ at 300 K. Reaction conditions: 10 mL ethanol; 2 µmol Pd; 4 mmol PhC≡CH; 303 K; 0.1 MPa H$_2$. Scanning parameters: **a** V$_s$ = −0.1 V, I = 1.0 nA; **d** V$_s$ = −0.2 V, I = 0.5 nA; **e** Vs = −1.0 V, I = 0.1 nA; **f** V$_s$ = 1.0 V, I = 0.1 nA.

be inert during the hydrogenation[41–43]. In situ temperature-programmed desorption–mass spectrometry (in situ TPD-MS) further confirmed that new species left on the used PdCu@Cu$_2$O catalyst were associated with deprotonated PhC≡C$^-$ species (Supplementary Figs. 28–31). DFT calculations demonstrated that PhC≡CH preferred to be adsorbed in a dissociative mode by forming the Cu(I)-C≡CPh structure at the defective sites of PdCu@Cu$_2$O (Supplementary Figs. 32–34). More importantly, the formation of Cu(I)-C≡CPh structure was able to effectively passivate the defective sites. DFT calculation results indicated that the surface Cu$_2$O overlayer with defects would be easily reduced by H$_2$ through a two-step reaction by overcoming a small barrier of 0.47 eV and 0.60 eV so that the whole Cu$_2$O overlayer could be quickly reduced with the "domino effect" (Supplementary Fig. 35). Our experiments did confirm that when there were no Cu(I)-C≡CPh motifs on the boundary, the surface Cu$_2$O of the fresh PdCu@Cu$_2$O catalyst would be easily reduced into metallic Cu by H$_2$ at 30 °C (Supplementary Fig. 36a, b). DFT calculations also revealed that the reduced Cu/Pd interface exhibited poor selectivity for the semi-hydrogenation of PhC≡CH due to the high hydrogenation activity for styrene (Fig. 3b, c and Supplementary Fig. 37), nicely explaining the catalysis results over the H$_2$-pretreated PdCu@Cu$_2$O catalyst. In contrast, the reduction of Cu$_2$O overlayer by H$_2$ was dramatically suppressed after the defective sites were "locked" by Cu(I)-C≡CPh groups as the removal of interfacial oxygen by H atom had to overcome a much high barrier of 1.85 eV (Supplementary Fig. 35). In addition, Cu(I)-C≡CPh motif would be stable upon hydrogenation due to a high barrier of 1.0 eV to be overcome (Supplementary Fig. 38), echoing the XPS measurements (Supplementary Fig. 36a, c).

All the above results clearly illustrated that PhC≡CH served as an unprecedented modifier for stabilizing catalytically active and selective sites for the semi-hydrogenation of alkynes. The selective semi-hydrogenation reactions should then take place over 2D Cu$_2$O/Pd interface. DFT calculations were further performed to explore the molecular mechanism of semi-hydrogenation of

PhC≡CH. The calculation results illustrated that H$_2$ would easily pass through the six-membered ring of the Cu$_2$O overlayer to be dissociated on the underneath Pd sites with an energy barrier of ~0.58 eV (Supplementary Fig. 39). The activated H atoms then migrated onto the O species at the Cu$_2$O and PdCu interface (Supplementary Fig. 40). Interestingly, the H atoms located on O-Cu(I) sites would not participate the hydrogenation, which was confirmed by nuclear magnetic resonance, in situ FT-IR, and isotope effect studies (Supplementary Figs. 41–44). The DFT calculations showed that the Cu$_2$O overlayer was involved in the semi-hydrogenation of PhC≡CH, and the electronic modulation by the PdCu alloy below Cu$_2$O overlayer would weaken the adsorption energy of H atoms, thus enhancing the subsequent hydrogenation (Fig. 3d and Supplementary Figs. 45–47, and Supplementary Table 4). The reaction barriers for the first and second H additions were calculated to 0.67 eV and 0.66 eV, respectively, lower than that of rate-determining step (TS1) over clean Pd(111) (0.84 eV) and Pd@Cu$_2$O (1.18 eV), explaining why the introduction of Cu helped enhance the hydrogenation activity of Pd. While the barriers of the following H addition step of styrene over 2D Cu$_2$O/Pd interface was 1.22 eV, much higher than the desorption energy of styrene (0.26 eV), accounting for the high selectivity of the catalyst towards styrene. On the contrary, over Pd(111), the desorption energy of styrene was higher than the barriers of hydrogenation (1.36 eV vs 0.92 eV, Supplementary Fig. 46), also consistent with the experimentally observed poor selectivity on unmodified Pd catalyst. The important role of surface microporous Cu$_2$O overlayer in the semi-hydrogenation of PhC≡CH was also confirmed by the catalytic performance depending on the alkyne size and the C≡C position (Supplementary Fig. 48).

**Semi-hydrogenation of a wide range of alkynes.** With the understanding of how to create 2D Cu$_2$O/Pd interface with enhanced activity, selectivity, and stability, we applied the

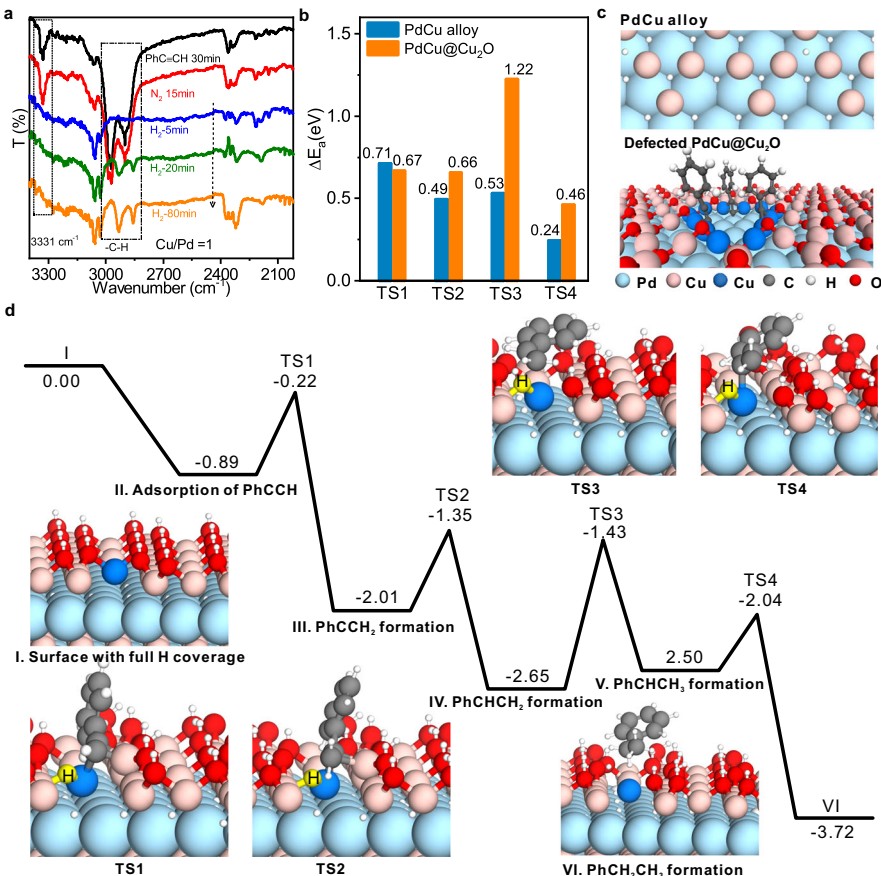

**Fig. 3 Mechanism of catalytic hydrogenation on 2D Cu₂O/Pd interface protected by Cu(I)-C≡CPh. a** Monitoring the catalytic hydrogenation of PhC≡CH on PdCu@Cu₂O using in situ FT-IR spectroscopy. **b** Predicted barriers of stepwise hydrogenation of PhC≡CH on the reduced PdCu alloy and PdCu@Cu₂O surface. **c** DFT-optimized structures of reduced PdCu alloy and defected PdCu@Cu₂O with dissociated PhC≡C⁻ bound on defect sites. **d** Energy profile of PhC≡CH stepwise hydrogenation on PdCu@Cu₂O surface protected by PhC≡CH from DFT calculations.

demonstrated concept to fabricate practical Pd catalysts for the semi-hydrogenation of a wide range of alkyne compounds. Following the protocol to grow Cu₂O overlayer on Pd NSs, an easily prepared Pd catalyst was made by reductive deposition of Cu on a Pd/C catalyst followed by air oxidation and pretreating with PhC≡CH (Supplementary Fig. 49). The performance of the as-prepared catalyst was evaluated for the semi-hydrogenation of a variety of alkynes (i.e. 4-tert-butylphenylacetylene, 4-ethynylanisole, 1-otyne, 3-hexyn-1-olcas, 1-phenyl-1-propyne, diphenylacetylene, 2-ethynyltoluene, and 2-ethynylanisole) which contain either terminal or internal C≡C groups. Compared with the untreated Pd/C, the modified catalyst did enhance both the activity and selectivity toward the alkene products (Table 1, entries 1–8, and Supplementary Figs. 50–53).

The selectivity to alkenes at 100% conversion was 96.3%, 95.4%, 95.1%, 95.7%, 96.0%, 95.2%, 94.8%, and 95.7% for 4-tert-butylstyrene, 4-methoxystyrene, 1-octene, cis-3-hexen-1-ol, cis-β-methylstyrene, stilbene, 2-vinyltoluene, and 1-ethenyl-2-methoxybenzene, respectively. In comparison, the unmodified Pd/C gave relatively poor selectivity of 85.4%, 84.4%, 71.9%, 74.5%, 86.9%, 76.7%, 58.5%, and 87.2%, respectively, under the same reaction conditions. Moreover, the modified catalyst also exhibited excellent catalytic performance in the hydrogenation of polar unsaturated compounds, such as 4-ethynylbenzaldehyde and 4-nitrophenylacetylene (Table 1, entries 9–10, and Supplementary Fig. 54). The deep understanding allows us to extend the interfacial catalytic sites far beyond the classical metal/support perimeter for fabricating

practical catalysts for the semi-hydrogenation of a wide range of alkynes under mild conditions.

In summary, the overgrowth of atomic-thick porous Cu₂O on Pd created an unprecedented 2D metal-support interface with remarkable performance enhancement towards the semi-hydrogenation of alkynes, far beyond the classical 1D metal/support perimeter. The comprehensive studies in this work have unexpectedly demonstrated three different roles of Cu in enhancing the catalysis of Pd: (1) The overgrowth of an atomic overlayer of 2D microporous Cu₂O on Pd is the key to enhancing the selectivity toward semi-hydrogenation due to the formation of 2D Cu₂O/Pd catalytic interface. (2) The alloy of Cu(0) with Pd is the main contributor to the enhanced activity due to the electronic modulation. (3) The formation of Cu(I)-alkyne complex via the easy reaction of the defective Cu₂O overlayer with terminal alkynes reinforces the robustness of the microporous Cu₂O overlayer on the catalyst surface, thus enhancing the stability of the catalytic interface against reduction during hydrogenation catalysis. The deep understanding of 2D metal-oxide interface not only enriched our knowledge and understanding of metal-support interaction but also provided a new direction to modulate the catalysis by encapsulating the active metals with porous 2D materials.

## Methods

**Synthesis of catalysts**. Pd(acac)₂ (50 mg), polyvinylpyrrolidone (160 mg), and tetrabutylammonium bromide (160 mg) were mixed with N,N-dimethylformamide (10 mL) and water (2 mL) in a glass pressure vessel. After being charged with CO to

**Table 1 Comparison of the catalytic performances of PdCu@Cu₂O-C≡CPh/C and references.**

| Entry | Substrate | Product | Pd/C | | PdCu@Cu₂O-C≡CPh/C | |
|---|---|---|---|---|---|---|
| | | | Conv. (%) | Sel. (%) | Conv. (%) | Sel. (%) |
| 1 | | | 100 | 85.4 | 99.6 | 96.3 |
| 2 | | | 97.5 | 84.4 | 98.6 | 95.4 |
| 3 | | | 100 | 71.9 | 100 | 95.1 |
| 4 | | | 64.1 | 74.5 | 100 | 95.7 |
| 5 | | | 99.8 | 86.9 | 100 | 96.0 |
| 6 | | | 68.9 | 76.7 | 98.8 | 95.2 |
| 7 | | | 95.9 | 58.5 | 100 | 94.8 |
| 8 | | | 99.3 | 87.2 | 100 | 95.7 |
| 9 | | | 78.5 | 76.8 | 99.7 | 97.6 |
| 10 | | | 65.1 | 78.4 | 100 | 96.9 |

Reaction conditions: 10 mL ethanol; 2 μmol Pd; 4 mmol terminal alkynes (or 1 mmol for other organics); $T = 303$ K; pressure = 0.1 MPa $H_2$.

1 bar, the vessel was heated from 30 °C to 60 °C for 30 min and kept at this temperature for 150 min with stirring. The resulting blue colloidal products were collected by centrifugation, and washed several times with ethanol and acetone for further use. For Pd@Cu core-shell NSs[22], the as-prepared Pd NSs were used as seeds and dispersed in 5 ml water in a 25 ml two-necked flask. The flask was heated at 60 °C, along with $H_2$ gas was continuously bubbled through the solution. After the system was kept at 60 °C for 40 min, the $H_2$ gas was moved away and the system remained to be closed. Some $Cu(NO_3)_2$ solution was injected by an injector under stirring. The ratio of Cu/Pd was varied from 0.25, 0.5, 0.75, 1, 1.25, 1.5, 2, and 3. The system was kept closed at 60 °C for 1 h. Then, the reaction was stopped by venting the system. To obtain PdCu@Cu₂O NSs, the as-prepared Pd@Cu catalysts were exposed to the air at 30 °C for 48 h.

**Catalytic tests**. The catalyst (Pd NSs, PdCu@Cu₂O, and Lindlar catalyst) was first dispersed in ethanol. A certain amount of dispersion was taken for the catalytic reaction, depending on the amount of catalyst needed. Typically, for PhC≡CH (or styrene) hydrogenation with a ratio of 1:2000 (Pd/PhC≡CH), 2 μmol Pd was used. The catalyst was dispersed in 10 ml ethanol in a glass pressure vessel and sonicated for 5 min, and then mixed with 4 mmol PhC≡CH under magnetic stirring. The vessel was then charged constantly with $H_2$ and kept at 30 °C in a water bath during stirring. The conversion and selectivity were characterized for the desired time period by extracting 100 μl solution for gas chromatographic analysis. For Cu nanocrystals, the same amount of Cu was also dispersed in ethanol for catalysis. The reaction conditions were kept the same as those for PdCu@Cu₂O. For the hydrogenation of other alkynes by the three (Pd/C, PdCu@Cu₂O-C≡CPh/C, or PdCu@Cu₂O/C) catalysts, the reaction condition was similar to that of PhC≡CH hydrogenation, but 4 mmol of other alkynes was used for the reaction. As a comparison, the catalysts were exposed to 1 bar $H_2$ for 30 min before PhC≡CH was introduced and the PhC≡CH hydrogenation reaction conditions were the same as above.

**STM experiments**. STM experiments were conducted in combined ultrahigh vacuum equipment consisting of a low-temperature STM chamber (LT-STM CreaTec) and a preparation chamber. LT-STM images were acquired at 78 K with a chemically etched W tip. All images were processed with SPIP software. Pd(111) single-crystal (Mateck) were cleaned by cycles of $Ar^+$ bombardment and annealing under UHV condition at 950 K, until no surface contaminant was detected by STM. PdCu surface alloy was constructed by evaporating >1.0 ML (monolayer) Cu on Pd(111) surface at 300 K, followed by annealing at 400 K for 10 min. Cuprous oxide was prepared by depositing Cu on Pd(111) or CuPd/Pd(111) surface in $1.0 \times 10^{-7}$ mbar $O_2$ at cryogenic temperature (~150 K) and then annealed at 400 K in oxygen atmosphere. PhC≡CH was purified by freeze-pump-thaw cycles, and introduced onto the sample surface through a leak valve at 300 K.

**Computational details**. All calculations were carried out by using the spin-polarized DFT embedding in the Vienna Ab Initio Simulation Package[44–47]. DFT + U with Perdew–Burke–Ernzerhof[48] exchange-correlation was employed. An U-J value of 6 eV for the Cu 3d states was applied, which has been used previously to investigate the Cu₂O[49]. The valence electrons were treated by plane-wave basis sets with a cutoff energy of 400 eV, while the core electrons were

described by projector augmented-wave pseudopotentials. The Monkhorst-Pack scheme was used to generate a k-point grid in the Brillouin zone, and the $3 \times 3 \times 1$ k-point grid was adopted for all calculations. The transition states were obtained by nudged elastic band method. And then the quasi-Newton algorithm was applied to refine the final transition state (TS) structures. For the geometric optimization, the forces of each ion were <0.03 eV/Å.

To simulate the PdCu@Cu$_2$O, Cu$_{12}$O$_8$ superstructure was place on a four-layer $(4 \times 4)$ Pd(111) slab with whole subsurface atoms displaced by Cu atoms, see Supplementary Fig. 20. During the optimization, the upper three layers (including the Cu$_{12}$O$_8$ layer) were allowed to be relaxed and the bottom two layers of Pd were frozen[50].

The adsorption energy $\Delta E_{ads}$ was defined as,

$$\Delta E_{ads} = E_{ads/surf} - E_{ads} - E_{surf} \qquad (1)$$

where $E_{ads/surf}$ represented the total energy of the catalyst and the adsorbed reactant molecules, $E_{ads}$ the energy of the reactant molecules, and $E_{surf}$ the energy of the clean surface of the catalyst.

The calculated surface energy ($\Delta E_{surf}$) of as-built model, was defined as,

$$\Delta E_{surf} = (E_{PdCu} - m \times E_{Pd} - n \times E_{Cu})/2A \qquad (2)$$

where $E_{PdCu}$, $E_{Pd}$, and $E_{Cu}$ denoted the energy of the PdCu alloys, the energy of one Pd atom in the bulk, and the energy of one Cu atom in the bulk, respectively. Here, m and n represented the number of Pd (or Cu) atoms in the alloy, respectively. The slab was treated with both of the exposing surfaces relaxed, and A is the area of one surface considered.

## Data availability

The data that support the findings of this study are available from the corresponding author upon reasonable request. Source data are provided in this paper.

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

## Acknowledgements

This work was supported by the National Key Research and Development Program of China (2017YFA0207302), the National Natural Science Foundation of China (21890752, 21731005, 22132004, 2213000173, 22072116, 92045303, 2212100020, 22102027) and the fundamental research funds for central universities (20720180026). N.Z. acknowledges support from the Tencent Foundation through the XPLORER PRIZE. K.L. thanks the China Postdoctoral Science Foundation Project (2020M682081). We also thank the beamline BL14W1 (Shanghai Synchrotron Radiation Facility) for providing beam time.

## Author contributions

N.Z. conceived and supervised the research project. K.L., L.J., and W.H. contributed equally to this work. N.Z. and K.L. designed and performed the synthesis and catalytic experiments. G.F. and L.J. designed and performed the DFT calculations. F.Y. and W.H. carried out the STM experiments. G.Z. performed the ACTEM characterizations. J.-F.L and Y.-J.Z. designed and carried out Raman measurements. C.X., R. Q., P.L., C.H, and J.W. were involved in part of the material characterizations. N.Z., G.F., F.Y., K.L., and L.J. prepared the manuscript. All the authors contributed to the overall scientific interpretation.

## Competing interests

The authors declare no competing interests.
