## [Peer Review File · Nature Communications]

Title: Atomic Overlayer of Permeable Microporous Cuprous Oxide on Palladium Promotes Hydrogenation CatalysisREVIEWER COMMENTS

Reviewer #1 (Remarks to the Author):

The manuscript by Liu et al. is not presented in a well-understandable way and it was difficult for me to follow. The language has to be improved. A few examples: on p.3: "scarce metals" (?), "...model catalysts ... were applied as model catalysts". The sentence on p. 5 begins with "And...". "The sample was pyrolysis little by little..." (in the Supp. Info). I could only guess that Fig. 2a refers to 0.45 ML coverage, while other (d-f) to 1.2 ML. It is also hard to figure out that Fig. 2c refers to powder catalysts and not to a single crystal system described in this figure. Below this figure (on p.7) the authors state: "It has been widely accepted (!) that the oxide overlayer should not fully cover the metal surface due lattice mismatch..." I do not understand how the film coverage correlates with the lattice mismatch.

As far as I understood, the main results can be summarized as follows. Pd nano-sheets (flakes) alloyed with Cu and then oxidized in air become more active in selective hydrogenation (i.e. alkyne to alkene and not to alkane). The catalyst pretreatment with alkyne (phenylacetylene) was beneficial as compared to that with pure H₂. The enhanced reactivity was assigned to the formation of a two-dimensional Cu₂O layer which is permeable for hydrogen and alkyne reacts on the Pd_{1-x}Cu_x(111) alloy surface underneath. Accordingly, the pretreatment effect was explained by that Cu-oxide reduction is suppressed by strongly bound alkynes on defect sites. To support the conclusions, the authors provide low-temperature STM images of the Pd(111) single crystal surface with an ultrathin Cu₂O layer grown on top, after several treatments in the ~ 10⁻⁶ mbar pressure range.

Although the topic of this study is interesting as it demonstrates the promotional effect of thin oxide layers on reactivity of metal surfaces, the conclusions are primarily based on theoretical calculations rather than solid experimental data.

It is unclear what was the thickness of the Pd nanosheets (2 nm as in ref. 21?), and also of the Cu-oxide film on Pd. For the latter ref. 23 says it is about 3-6 layers. If so, I wonder how such a "thick" film is permeable for alkynes to reach the metal surface. For STM studies, the Cu₂O surface was prepared at UHV compatible pressures, and the treatments were also done at low pressures. I wonder whether a well-defined Cu-oxide layer used for calculations remains under H₂-rich reaction conditions. The formation of Cu⁺ in the post-reacted samples could be explained by oxidation of Cu-Pd alloy in air.

Figure 2c highlights the pretreatment effect (either H₂ or alkyne prior to the reaction) but does not show the results for "normal" reaction when both reactants are exposed simultaneously. I anticipate the result to be similar to that with alkyne pre-adsorption. If so, I would turn this other way around: the pretreatment with pure H₂ reduces and eventually destroys the Cu-oxide layer and hence the active sites rather than alkyne "serves as an unprecedented modifier" (p. 10).

As to other alkynes tested, should the reactivity also depend on the alkyne size and triple bond position to adsorb on the metal surface in the "pore"?

What is the “electronic modulation” of the Pd-Cu surface the author talk about in the abstract?

Overall, I think the hypothesis sounds interesting, but at present it is solely based on DFT calculations rather than experiment.

Reviewer #2 (Remarks to the Author):

This paper investigates a well-defined system of Pd nanocrystals decorated with Cu overlayer for the semihydrogenation of aryl alkynes such as phenylacetylene. By altering the ratio of Cu/Pd, the authors identify an important change in the speciation of Cu, as quantities of Cu below a monolayer ($\text{Cu/Pd} < 0.5$) prefer to migrate subsurface and above a monolayer ($\text{Cu/Pd} > 0.5$), excess Cu is left on the Pd surface to form Cu_2O phase after exposure to air. The formation of some Cu_2O phase is identified as being crucial to the highly selective, active, and stable conversion of phenylacetylene to styrene over $\text{PdCu@Cu}_2\text{O}$. Furthermore, the authors identify that a secondary catalyst effect responsible for the increased stability of Cu_2O even in reducing atmospheres is the stabilization of Cu_2O phase by dissociatively adsorbed phenylacetylene, which locks in the Cu_2O phase by complexing with Cu(I) and increasing the barrier for reduction, thus enabling the active interface between Cu_2O and Pd to be stable under reaction conditions. The work follows a very detailed and logical progression and highlights a unique and powerful example of substrate-modified catalyst behavior, which is not often understood. This paper should be published in Nature Communications after a few issues are clarified.

1. The proposed mechanism involves facile adsorption of H_2 onto Pd below the Cu_2O overlayer due to the large diameter of the Cu_2O pore size (5.5 Å). However, in similar cases of SMSI using other oxides, small molecule adsorption (CO or H_2) is often cited to be highly suppressed. It would be interesting to see what the H_2 uptake of $\text{PdCu@Cu}_2\text{O}$ catalysts were for comparison against similar SMSI catalysts to understand the role of Cu_2O porosity. Furthermore, this study would be even more illuminating after phenylacetylene pre-treatment, as H_2 uptake after phenylacetylene exposure would give insight into whether or not these two species can be co-adsorbed in significant quantities, or if phenylacetylene effectively blocks H_2 adsorption. Such site blockage by phenylacetylene might have important impacts on the observed selectivity as well.
2. The reduction of Cu_2O at 30°C is attributed solely to defective sites on the Cu_2O lattice, but the possibility of spillover from Pd domains which are not decorated (as evidence by EDS, etc.) which should near-barrierlessly activate H_2 and thus facilitate Cu_2O reduction even at low temperatures is not explored or discussed. The presence of these defective Cu_2O sites is also only assumed and used frequently in theoretical models, but their presence is not necessarily established by any of the characterization. Considering spillover for facilitated reduction is well-documented on multiple systems involving well-mixed noble metals/metal oxides, this may play a larger role than what is addressed here by the authors.

3. Citations 41-43 are works which also investigated dissociative phenylacetylene adsorption, sometimes on Cu₂O catalysts, using FT-IR. However, in none of those papers is an IR active mode for Cu:CCPh identified. Citation 43 reports that there is a loss of both the alkyne and alkynal C-H vibration mode upon terminal adsorption. These authors identify the same loss in alkynal C-H, but make an identification of a Cu:CCPh mode which has not been previously been identified, at least in the citations given. This may suggest it is either not associated with the Cu:CCPh complex, or that there is a different geometric bonding mode which makes this vibration IR active.

4. The Cu XPS spectra in Sup. Fig. 36a are not labelled clearly enough to distinguish that there is any change to the materials. The issue is that the energy axis scale is so large compared to the magnitude of the shift that it's hard to visually see the shift which is important to follow the argument being verbally made by the authors. Perhaps including numbers (like in Sup. Fig. 6a) above the Cu XPS signals to highlight that they are shifted or make use a separate zoom where the shifts or changes to the signals are more noticeable.

Response to Reviewer #1

1. **General Comment:** The manuscript by Liu et al. is not presented in a well-understandable way and it was difficult for me to follow. The language has to be improved. A few examples: on p.3: “scarce metals” (?), “...model catalysts ... were applied as model catalysts”. The sentence on p. 5 begins with “And...”. “The sample was pyrolysis little by little...” (in the Supp. Info). I could only guess that Fig. 2a refers to 0.45 ML coverage, while other (d-f) to 1.2 ML. It is also hard to figure out that Fig. 2c refers to powder catalysts and not to a single crystal system described in this figure.

Response: Thank the reviewer for pointing out the language problem. Following the suggestion, we have tried our best to revise the manuscript accordingly so that it is presented more clearly.

2. **Comment:** Below this figure (on p.7) the authors state: “It has been widely accepted (!) that the oxide overlayer should not fully cover the metal surface due lattice mismatch....” I do not understand how the film coverage correlates with the lattice mismatch.

Response: What we want to state is that the lattice mismatch between oxide overlayer and metal makes it challenging to grow atomic oxide overlayer that fully covers the metal surface. There should be defective sites on the oxide overlayer. Due to strain-induced reduction of the vacancy formation energy, the observation of the growth of defective oxide overlayer on metal is quite common (*J. Phys. Chem. Lett.* **2016**, *7*, 1303-1309; *Phys. Rev. Lett.* **1997**, *79*, 4222-4225; *Nano Res.* **2018**, *11*, 5957-5967; *J. Phys. Chem. C* **2019**, *123*, 12716–12721; *Surf. Sci.* **2004**, *554*, L120–L126; *Nat. Commun.* **2017**, *8*, 14039).

3. **Comment:** As far as I understood, the main results can be summarized as follows. Pd nano-sheets (flakes) alloyed with Cu and then oxidized in air become more active in selective hydrogenation (i.e. alkyne to alkene and not to alkane). The catalyst pretreatment with alkyne (phenylacetylene) was beneficial as compared to that with pure H₂. The enhanced reactivity was assigned to the formation of a two-dimensional Cu₂O layer which is permeable for hydrogen and alkyne reacts on the Pd_{1-x}Cu_x(111) alloy surface underneath. Accordingly, the pretreatment effect was explained by that Cu-oxide reduction is suppressed by strongly bound alkynes on defect sites. To support the

conclusions, the authors provide low-temperature STM images of the Pd(111) single crystal surface with an ultrathin Cu₂O layer grown on top, after several treatments in the ~ 10⁻⁶ mbar pressure range. Although the topic of this study is interesting as it demonstrates the promotional effect of thin oxide layers on reactivity of metal surfaces, the conclusions are primarily based on theoretical calculations rather than solid experimental data.

Response: Thank the reviewer very much for the recognition of the contribution of our work. As described in the manuscript, in addition to the DFT analysis, *in situ* FT-IR, TPD-MS, XPS, STM analysis, and detail control experiments were conducted to explore the microstructure of PdCu@Cu₂O under different atmosphere as well as the molecular mechanism of semi-hydrogenation of PhC≡CH catalyzed by PdCu@Cu₂O.

- 1) As shown in **Supp. Figs. 3-6**, there were two kinds of Cu coexist in the PdCu@Cu₂O. While the Cu underneath the Pd surface preferred to keep the metallic state (Cu⁰), the surface Cu would be oxidized into +1 (Cu⁺).
- 2) As shown in **Fig. 1e, 2c** and **Supp. Figs. 8 to 10, 22, 36**, the difference in catalytic performance and Cu LMM XAES spectra confirmed the defective Cu₂O on PdCu@Cu₂O was easily reduced to Cu in H₂ at 30°C. Once Cu₂O was reduced, the catalyst lost its semi-hydrogenation selectivity. Thus, the semi-hydrogenation reactions should take place over 2D Cu₂O/Pd interface stabilized by alkynes.
- 3) As shown in **Fig. 3a** and **Supp. Figs. 26 to 31, 36**, the reduction of Cu₂O overlayer by H₂ was dramatically suppressed after the defective sites were “locked” by Cu(I)-C≡CPh groups, indicating that PhC≡CH served as an unprecedented modifier for stabilizing 2D Cu₂O/Pd catalytic interface for the semi-hydrogenation of alkynes.
- 4) As shown in **Fig. 3a** and **Supp. Fig. 27**, the PhC≡CH that participate the semi-hydrogenation was in molecular form.
- 5) Nuclear magnetic resonance (NMR) and *in situ* FT-IR analysis confirmed that the Cu(I)-O-H would not participate the semi-hydrogenation (**Supp. Figs. 42-44**).
- 6) Once PhC≡CH was introduced, PdCu@Cu₂O would exhibit negligible activity for hydrogenation of styrene (**Supp. Fig. 26**) but excellent selectivity for the semi-hydrogenation of a wide range of alkyne compounds (**Table 1 and Sup. Figs. 48-54**).

With all these results, we consider that the proposed mechanism was well supported by both experiments and calculations.

4. **Comment:** It is unclear what was the thickness of the Pd nanosheets (2 nm as in ref. 21?), and also of the Cu-oxide film on Pd. For the latter ref. 23 says it is about 3-6 layers. If so, I wonder how such a “thick” film is permeable for alkynes to reach the metal surface.

Response: Based on the previous work of our group (ref. 21), the thickness of the Pd nanosheets remained unchanged (1.8 nm, 5 atomic layers) during the reaction. For the Cu-oxide film on Pd, the optimal ratio of Cu/Pd to achieve the best catalytic activity was 1:1. Since part of Cu was diffused into Pd to form PdCu, and Cu₂O overlayer should cover both the upper and lower surface of Pd nanosheets, the surface Cu₂O overlayer on PdCu alloy would not be too thick to prevent the access of alkynes. In ref. 23, Cu₂O overlayer with 3- to 6-layer thickness were achieved by using higher molar ratio of Cu/Ag, varied from 2:1 to 10:1.

5. **Comment:** For STM studies, the Cu₂O surface was prepared at UHV compatible pressures, and the treatments were also done at low pressures. I wonder whether a well-defined Cu-oxide layer used for calculations remains under H₂-rich reaction conditions. The formation of Cu⁺ in the post-reacted samples could be explained by oxidation of Cu-Pd alloy in air.

Response: Our experiment results and DFT analysis confirmed that the surface defective Cu₂O overlayer of the fresh PdCu@Cu₂O catalyst would be easily reduced into metallic Cu by H₂ (H₂-rich) at 30 °C (**Supp. Fig. 36a, b**). However, when there were Cu(I)-C≡CPh motifs at the defective sites of surface Cu₂O, the reduction of Cu₂O overlayer by H₂ was dramatically suppressed because the removal of interfacial oxygen by H atom has to overcome a much high barrier of 1.85 eV (**Supp. Figs. 35-36**). All the above results clearly illustrated that Cu(I)-C≡CPh served as an unprecedented modifier for stabilizing surface Cu₂O and 2D Cu₂O/Pd interface for the semi-hydrogenation of alkynes. The electronic structure and coordination environment of PdCu@Cu₂O remained unchanged during the reaction. Thus, the formation of Cu⁺ in the post-reacted samples should not be caused by the oxidation of Cu-Pd alloy in air.

6. **Comment:** Figure 2c highlights the pretreatment effect (either H₂ or alkyne prior to the reaction) but does not show the results for “normal” reaction when both reactants are exposed simultaneously. I anticipate the result to be similar to that with alkyne pre-adsorption. If so, I would turn this other way around: the pre-treatment with pure H₂ reduces and eventually destroys the Cu-oxide layer and hence the active sites rather than alkyne “serves as an unprecedented modifier” (p. 10).

Response: Following the reviewer’s comment, we further considered the catalytic performance of PdCu@Cu₂O for “normal” reaction when H₂ and PhC≡CH are exposed simultaneously. As shown in **Fig. R1**, when the conversion of PhC≡CH reached 100%, the selectivity toward styrene was only 91.5% for “normal” reaction, which was much lower than that of PhC≡CH pretreatment (96.9%). Moreover, by extending the reaction time, the selectivity towards styrene would gradually decay for “normal” reaction, different from that of PhC≡CH-pretreated PdCu@Cu₂O catalyst. These results suggested the formation of Cu-C≡CPh structure should be suppressed in the presence of H₂ at “normal” reaction.

Figure R1. The difference in catalytic performance of PdCu@Cu₂O (Cu/Pd =1) caused by the feeding sequence. Reaction conditions: 10 mL ethanol; 2 μmol Pd; 4 mmol PhC≡CH; 303 K; 0.1 MPa H₂.

7. **Comment:** As to other alkynes tested, should the reactivity also depend on the alkyne size and triple bond position to adsorb on the metal surface in the “pore”?

Response: Yes, the catalytic performance of PdCu@Cu₂O also depends on

the alkyne size and triple bond position due to the different adsorption on the metal surface in “pore”. To further test the influence of the alkyne size, acetylene and propyne were chosen as the model reagents. As shown in **Fig. R2a**, total acetylene conversion was achieved at 140 °C over PdCu@Cu₂O with the ethylene selectivity of only 57.6%. In contrast, when the conversion of propyne reached 100% at 160 °C, the selectivity of propylene was still as high as 94.2%. PdCu@Cu₂O exhibited much better performance in the semi-hydrogenation of propyne than acetylene, indicating that the catalytic performance of PdCu@Cu₂O depends on the alkyne size.

To further test the influence of steric effect, 2-ethynyltoluene and its positional isomers were chosen as the model reagents. As shown in **Fig. R2b**, the reactivity was increased from ortho to meta, to para arrangements, and the highest activity was achieved for para-substituted toluene, indicating the “pore” at the PdCu@Cu₂O interface would exhibit certain shape selectivity.

Figure R2. (a) Gas phase hydrogenation of alkyne in a large excess of alkene over PdCu@Cu₂O (Cu/Pd=1). Reaction condition: a gas mixture with a space velocity of 60,000 ml h⁻¹g⁻¹ was introduced into the reactor, simulating the front-end hydrogenation conditions with 1.0 vol.% C₂H₂ (or C₃H₄), 20.0 vol.% H₂ and 20.0 vol.% C₂H₄ (or C₃H₆), balanced with N₂. (b) The difference in catalytic performance of PdCu@Cu₂O (Cu/Pd =1) caused by the triple bond position. Reaction conditions: 10 mL ethanol; 2 μmol Pd; 4 mmol alkyne; 303 K; 0.1 MPa H₂.

8. **Comment:** What is the “electronic modulation” of the Pd-Cu surface the

author talks about in the abstract?

Response: Part of copper is diffused into palladium lattice to form near surface Pd-Cu alloy whose electronic structure would influence the binding of activated H atoms on metal. The alloyed Pd-Cu surface exhibited weak adsorption of dissociated H atoms so that the reactivity of the hydrogenation activity was enhanced. To illustrate the “electronic modulation” of the Pd-Cu alloy, additional calculations on Pd@Cu₂O were conducted, and the energy barrier of TS1'' was predicted to be 1.18 eV (**Fig. R3**), much higher than that of PdCu@Cu₂O (0.67 eV).

Figure R3. Energy profile and optimized structures of the TSs for stepwise hydrogenation of PhC≡CH on the Pd@Cu₂O surface .

Response to Reviewer #2

1. **Comment:** The proposed mechanism involves facile adsorption of H₂ onto Pd below the Cu₂O overlayer due to the large diameter of the Cu₂O pore size (5.5 Å). However, in similar cases of SMSI using other oxides, small molecule adsorption (CO or H₂) is often cited to be highly suppressed. It would be interesting to see what the H₂ uptake of PdCu@Cu₂O catalysts were for comparison against similar SMSI catalysts to understand the role of Cu₂O porosity.

Response: Following the reviewer's suggestion, we compared the adsorption energies of CO and H₂ on different surfaces/interfaces, see **Figure R4** and **Table R1**. For CO adsorption, the predicted adsorption energies increased from Pd(111) (-2.07 eV), to PdCu (111) (-1.83 eV), to PdCu@Cu₂O (-0.95 eV). And, the same tendency can be found for dissociated adsorption of H₂, in which the adsorption energies for the two H atoms were predicted to be -1.03 eV, -0.87 eV and -0.21 eV for Pd(111), PdCu (111) and PdCu@Cu₂O, respectively. All these findings indicated that the Cu₂O overlayer does suppressed the adsorption of both CO and H₂, in line with the expectation from SMSI effect.

Figure R4. Structures of dissociated H atoms and CO molecule adsorbed on

Pd(111), PdCu (111) alloy, and PdCu@Cu₂O surfaces (a-l).

Table R1. The adsorption energies of the dissociated H atoms and the CO molecule on Pd(111), PdCu (111), and PdCu@Cu₂O surfaces.

Surfaces	$\Delta E_{\text{ads}}(2\text{H})/\text{eV}$	$\Delta E_{\text{ads}}(\text{CO})/\text{eV}$
Pd(111)	-1.03	-2.07
PdCu(111)	-0.87	-1.83
PdCu@Cu ₂ O	-0.21	-0.95

2. Comment: Furthermore, this study would be even more illuminating after phenylacetylene pre-treatment, as H₂ uptake after phenylacetylene exposure would give insight into whether or not these two species can be co-adsorbed in significant quantities, or if phenylacetylene effectively blocks H₂ adsorption. Such site blockage by phenylacetylene might have important impacts on the observed selectivity as well.

Response: Thank the reviewer for the comment. As described in the manuscript, the PhC≡CH pre-treatment plays a vital role in stabilizing the 2D Cu₂O/Pd interface, and does not block the adsorption and activation of H₂ on PdCu@Cu₂O. H₂-D₂ exchange was employed to further characterize the H₂ dissociation activity over the PhC≡CH pre-treated PdCu@Cu₂O catalyst at ambient pressure. As shown in **Fig. R5**, HD was produced right after H₂ and D₂ were co-introduced to the PhC≡CH pre-treated PdCu@Cu₂O, indicating the active sites were still accessible for H₂ even with protection of PhC≡CH.

Figure R5. The isothermal H₂-D₂ exchange over (a) the PhC≡CH-pretreated PdCu@Cu₂O catalyst and (b) blank at 30 °C.

3. **Comment:** The reduction of Cu₂O at 30 °C is attributed solely to defective sites on the Cu₂O lattice, but the possibility of spillover from Pd domains which are not decorated (as evidence by EDS, etc.) which should near-barrierlessly activate H₂ and thus facilitate Cu₂O reduction even at low temperatures is not explored or discussed. The presence of these defective Cu₂O sites is also only assumed and used frequently in theoretical models, but their presence is not necessarily established by any of the characterization. Considering spillover for facilitated reduction is well-documented on multiple systems involving well-mixed noble metals/metal oxides, this may play a larger role than what is addressed here by the authors.

Response: We appreciate the constructive and detailed comments from the reviewer. The defective structure of Cu₂O overlayer was related to the lattice mismatch between Cu₂O(111) and PdCu alloy below the Cu₂O overlayer, which was further confirmed by STM and electrochemical analysis (**Sup. Figs. 4, 14, 19, and 23**). We agree with the reviewer that the H atoms activated on the Pd domains would easily spill over the Cu₂O overlayer. In fact, we have demonstrated the surface Cu₂O of the fresh PdCu@Cu₂O catalyst would be easily reduced into metallic Cu by H₂ at 30 °C. And the reduction of surface Cu₂O can be further extended from the defective sites to the whole surface (**Fig. 2d-f and Sup. Figs. 23, 35-36**). The reduced Cu/Pd interface exhibited poor selectivity for the semi-hydrogenation of PhC≡CH (**Fig. 2c, 3b, c and Sup. Fig. 37**), clearly indicating that the spilled H from Pd domains to the defective Cu₂O does not contribute to the high selectivity. Furthermore, the reduction of Cu₂O overlayer by H₂ would be dramatically suppressed after the defective sites were “locked” by Cu(I)-C≡CPh groups (**Sup. Figs. 35 and 36**). Therefore, the 2D Cu₂O/Pd catalytic interface for the semi-hydrogenation of alkynes should be protected by Cu(I)-C≡CPh groups before exposed to H₂.

4. **Comment:** Citations 41-43 are works which also investigated dissociative phenylacetylene adsorption, sometimes on Cu₂O catalysts, using FT-IR. However, in none of those papers is an IR active mode for Cu:CCPh identified. Citation 43 reports that there is a loss of both the alkyne and alkynal C-H vibration mode upon terminal adsorption. These authors identify the same loss in alkynal C-H, but make an identification of a Cu:CCPh mode which has not been previously been identified, at least in the citations given. This may suggest

it is either not associated with the Cu:CCPh complex, or that there is a different geometric bonding mode which makes this vibration IR active.

Response: Thank the reviewer for the comment. *In situ* TPD-MS characterization confirmed that the new species on PdCu@Cu₂O-used was the dissociated PhC≡C-, and the dissociated PhC≡C- was related to Cu (**Sup. Figs. 28-31**). The configuration of Cu(I)-C≡CPh structure can be identified by comparing *in situ* FT-IR spectra. The alkynyl C-H stretching peaks of the free ligands at ~3300 cm⁻¹ disappear upon adsorption, indicating that the alkynyl ligands (PhC≡C-) should be bound to PdCu@Cu₂O (**type 1 or 2**) (**Fig. 3a** and **Sup. Fig. 27**). In principle, the non-polar group -C≡C- stretching vibration is IR inactive while Raman active. Thus, we also used surface-enhanced Raman scattering (SERS) spectra to detect the vibration of -C≡C- stretching. It was observed that -C≡C- stretching is red-shifted from 2110 cm⁻¹ to 1971 cm⁻¹ (**Sup. Fig. 27b**), suggesting that the alkynyl group would be bonded in an upright configuration (**type 1**) rather than in a flat-lying configuration (*J. Am. Chem. Soc.* **2013**, *135*, 9450–9457). Overall, the formation and configuration of Cu(I)-C≡CPh structure were confirmed by *in situ* TPD-MS, *in situ* FT-IR, and SERS.

5. **Comment:** The Cu XPS spectra in Sup. Fig. 36a are not labelled clearly enough to distinguish that there is any change to the materials. The issue is that the energy axis scale is so large compared to the magnitude of the shift that it's hard to visually see the shift which is important to follow the argument being verbally made by the authors. Perhaps including numbers (like in Sup. Fig. 6a) above the Cu XPS signals to highlight that they are shifted or make use a separate zoom where the shifts or changes to the signals are more noticeable.

Response: Thanks for the suggestion. Following the suggestion, the Cu XPS spectra in **Sup. Fig. 36a** has been revised to highlight the shift.

REVIEWERS' COMMENTS

Reviewer #2 (Remarks to the Author):

The authors reasonably addressed the referee concerns. The paper can be published now.

Response to Reviewer #2

Comment: The authors reasonably addressed the referee concerns. The paper can be published now.

Response: We thank the reviewer for the publication recommendation.